

# Identification of two short peptide motifs from serine/arginine-rich protein ribonucleic acid recognition motif-1 domain acting as splicing regulators

Tao Jiang[1,2,*], Li Wang[1,*], Liang Tang[1], Azhar Zeb[1] and Yanjun Hou[1]

[1] Jiangsu Key Laboratory for Molecular and Medical Biotechnology, College of Life Sciences, Nanjing Normal University, NanJing, China
[2] Department of Rehabilitation, Southwest Hospital, Third Military Medical University Army Medical University, Chongqing, China
* These authors contributed equally to this work.

## ABSTRACT

**Background:** Serine/arginine-rich (SR) proteins regulate pre-mRNA splicing. However, structurally similar proteins often behave differently in splicing regulation and the underlying mechanisms are largely unknown. Here, using *SMN1/2* minigenes we extensively analyzed four SR proteins, SRSF1/5/6/9.

**Methods:** In this study, the effects of these proteins on *SMN1/2* exon 7 splicing when tethered at either intron 6 or 7 were evaluated using an MS2-tethering assay. Deletion analysis in four SR proteins and co-overexpression analysis were performed.

**Results:** Splicing outcomes varied among all four SR proteins, SRSF1 and SRSF5 function the same at the two sites, acting as repressor and stimulator, respectively; while SRSF6 and SRSF9 promote exon 7 inclusion at only one site. Further, the key domains of each SR proteins were investigated, which identified a potent inhibitory nonapeptide in the C-terminus of SRSF1/9 ribonucleic acid recognition motif-1 (RRM1) and a potent stimulatory heptapeptide at the N-terminus of SRSF5/6 RRM1.

**Conclusion:** The insight of the four SR proteins and their domains in affecting *SMN* gene splicing brings a new perspective on the modes of action of SR proteins; and the functional peptides obtained here offers new ideas for developing splice switching-related therapies.

## INTRODUCTION

Pre-mRNA splicing is a fundamental step of gene expression, in which high specificity and fidelity in the recognition of splice sites by the spliceosome is required to ensure protein production accuracy (*Lee & Rio, 2015*). Aside from the core splice signals (the 5′ and 3′ splice sites, polypyrimidine tract and branchpoint site), *cis* splicing regulatory elements (SREs), including exonic/intronic splicing enhancers/silencers (ESE/ISE & ESS/ISS), also play crucial roles in the process (*Re et al., 2014*; *Cartegni, Chew & Krainer, 2002*). Serine/arginine-rich (SR) proteins are a family of important regulatory RNA-binding proteins (RBPs) that consist of one or two highly conserved RNA-recognition motifs (RRM1/2) and

Corresponding authors
Li Wang, wangli_nnu@njnu.edu.cn
Yanjun Hou, houy@njnu.edu.cn

an arginine/serine-rich (RS) domain (*Zahler et al., 1992*). One of their roles is to regulate pre-mRNA splicing through binding to specific SREs (*Krainer, Conway & Kozak, 1990*). In humans, a total of 12 proteins belong to the family, named SRSF1-12, respectively (*Manley & Krainer, 2010*). They not only regulate constitutive splicing but also play a more complex role in alternative splicing regulation (*Jeong, 2017*). Additionally, they are involved in other mRNA metabolism processes, such as mRNA extranuclear transport, nonsense mediated mRNA decay and mRNA translation (*Krainer, Conway & Kozak, 1990*; *Long & Caceres, 2009*). Deregulation of SR proteins has been found to be highly associated with cancer (*Das & Krainer, 2014*).

SR proteins participate in the early stages of splice-site recognition and spliceosome assembly as well as in the two-step splicing process (*Zhu & Krainer, 2000*). They have been well recognized as splicing facilitators under two known models. In the "recruitment model", an SR protein bound to an ESE facilitates the interactions between the spliceosome and splice sites. In the "inhibitor model", an SR protein functions by antagonizing an ESS-bound heteronuclear RNP (hnRNP) protein (*Long & Caceres, 2009*). SR proteins also form a protein-protein interaction network between introns, which juxtaposes the 5′ and 3′ splice sites at the early stage of spliceosome assembly, or directly interacts with the branchpoint to promote splicing (*Shen, Kan & Green, 2004*; *Shen & Green, 2004*). Furthermore, SR proteins may act as splicing inhibitors. For example, SRSF1 and SRSF5 interfere with splice-site recognition to promote exon skipping through binding to an ISS of *CFTR* (*Buratti et al., 2007*). In many alternative splicing cases, different SR proteins display distinct effects. In a study on *CD44* pre-mRNA splicing of variant exon 6 (V6), SRSF3 and SRSF4 had no effect on V6 splicing, whereas SRSF1, SRSF6 and SRSF9 significantly decreased V6 levels by inhibiting exon 6 inclusion (*Loh et al., 2016*). SRSF1 and SRSF6 have been shown to function oppositely when bound to upstream (in exon) *vs.* bound downstream (in intron) of the 5′ splice site in an reporter minigene (*Erkelenz et al., 2013*).

The protein structure of SRSF1 (SF2/ASF) has been extensively investigated to understand the roles of each domain in splicing regulation. Basically, its RRM(s) are responsible for direct binding to SREs (*Moon et al., 2019*). Recently, the canonical RRM (RRM1) of SRSF1 have been found to preferentially bind to a CN (N represents any nucleotide) motif (*Clery et al., 2021*). The pseudo-RRM (RRM2) of SRSF1 recognizes a GGA motif with the interaction centered on the α1-helix of the RRM (*Clery et al., 2013*). RS domains are thought to function in recruiting components of the basal splicing machinery through protein-protein interactions (*Phelan et al., 2012*; *Hertel & Graveley, 2005*). However, splicing can be rescued by SR proteins in an RS domain-independent manner, indicating that RRMs can perform functions other than RNA binding in certain circumstances (*Zhang et al., 2014*). In cell-free S-100 complementation assays, *Zhu & Krainer (2000)* showed that the RS domain of SRSF2 is dispensable for constitutively spliced substrates or some substrates with ESEs. SRSF1 lacking its RS domain also stimulated splicing of HIV *tat* intron 2 through binding to an ESE in exon 3 to antagonize a downstream ESS (*Tange & Kjems, 2001*). In another study, SRSF1, in the absence of RS domain, promoted splicing of an IgM pre-mRNA substrate when the inhibitory

N-terminal amino acids were deleted (*Shaw et al., 2007*). *Cho et al. (2011)* revealed that both RRM1 and RRM2 directly contact U1-70K, a spliceosomal component through RRM-RRM interaction, and thus promote U1 annealing to the 5′ splice site.

Our understanding on the functions of SR proteins and their domains remains limited. In particular, why SR proteins with high structural similarity exert different or even opposite effects on splicing in many alternative splicing events has not been well elucidated. In this study, we used the *survival of motor neuron 1 and 2* (*SMN1/2*) genes as model genes to compare four structurally similar SR proteins and dissect the roles of their key domains. *SMN1/2* are two paralogous genes associated with spinal muscular atrophy (SMA) and the splicing patterns of their exon 7 have been well characterized (*Wu et al., 2017*; *Hua et al., 2007*; *Palacino et al., 2015*). The *SMN2* gene, a nearly identical copy of *SMN1* (>99.9% sequence identity), fails to rescue SMA due to a single nucleotide mutation of exon 7 (*Monani et al., 1999*; *Lorson et al., 1999*). This mutation alters *SMN2* splicing pattern, resulting in a strong inhibition of exon 7, making these two genes ideal models for studying pre-mRNA alternative splicing.

In this study, we examined four SR proteins (SRSF1, SRSF5, SRSF6, and SRSF9); all of them consist of two RRM domains and one RS domain. Using an MS2-tethering assay, exogenously expressed SR fusion proteins and their deletion mutants were tethered to either intron 6 (position −36) or intron 7 (position +37) to explore their effects on splicing. SRSF1 inhibited *SMN1/2* exon 7 splicing bound at both sites, whereas SRSF5 promoted *SMN1/2* exon 7 splicing at both sites. SRSF6 and SRSF9 displayed position-dependent effects on *SMN1/2* exon 7 splicing. Domain analyses revealed that RRM domains generally possessed position preference which promoted splicing when tethered to intron 6 but inhibited splicing when tethered to intron 7. In addition, we identified a conserved heptapeptide motif in the N-terminus of SRSF5/6 RRM1 that significantly promoted *SMN1/2* exon 7 splicing, and a nonapeptide motif in the C-terminus of SRSF1/9 RRM1 that inhibited *SMN1/2* exon 7 splicing. Our data shed light on why structurally similar SR proteins cause different outcomes in splicing regulation.

## MATERIALS AND METHODS

### Plasmid construction

The pCl-*SMN1/2* minigene constructs were the same as previously described (*Hua et al., 2007*). The two minigenes comprise the 111-nt exon 6, a truncated 200-nt intron 6, the 54-nt exon 7, the 444-nt intron 7, and the first 75 nt of exon 8, followed by a 9 nt consensus 5′ splice site (CAGGTAAGT). Plasmid pCl-SMN-MS2-In6 was generated by replacing the sequence from −47 to −36 position of exon 7 with a 19-nt MS2 hairpin sequence (5′-ACATGAGGATCACCCATGT-3′) (*Wu et al., 2017*). The MS2 binding sequence (5′-GCGTACACCATCAGGGTACGC-3′) was inserted into the +37 position of the *SMN1/2* intron 7 by site-directed mutagenesis to form pCl-SMN-MS2-In7 (*Sun et al., 2012*). All expression plasmids were constructed with vector pCGT7, and thus all expressed proteins have an N-terminal T7 tag. Plasmids pCGT7-(CP)-SRSF and SRSF deletion mutants is generated by using restriction sites and/or SLIC methods. Primer pairs used for cloning are listed in Table S1.

## Cell culture and transfection

HEK293 cells (National Collection of Authenticated Cell Cultures, China) were cultured in Dulbecco's modified Eagle's medium (DMEM; Invitrogen, Carlsbad, CA, USA) supplemented with 10% (v/v) fetal bovine serum (FBS) and antibiotics (100 U/ml penicillin and 100 mg/ml streptomycin) at 37 °C in a 5% $CO_2$ atmosphere. For splicing analysis of SR protein on *SMN1/2* minigenes, $10^5$ cells per well were seeded in 12-well plates in DMEM with 10% FBS. Data were collected as previously described in *Wu et al. (2017)*. The next day, 500 ng of each minigene plasmid together with or without 300 ng of each protein-expression plasmid was delivered to cells using branched polyethylenimine reagent (Sigma-Aldrich, St. Louis, MO, USA).

## Fluorescence-labelled RT-PCR

Cells were harvested 36-h post-transfection, and total RNA was isolated with TRizol reagent (Vazyme, Nanjing, China); 700 ng of each RNA sample was used per 10 μL reaction for first-strand cDNA synthesis with oligo (dT) 18 and M-MLV reverse transcriptase (Vazyme, Nanjing, China). For minigenes, splicing products were amplified semi-quantitatively using 26 cycles (95 °C for 15 s, 60 °C for 30 s, and 72 °C for 25 s) with forward primer T7-F (5′-TACTTAATACGACTCACTATAGGCTAGCCTCG-3′) and Cy5-labeled reverse primer Ex8-29to52-R (5′-Cy5-TCTGATCGTTTCTTTAGTGGTGTC-3′), as previously described (*Gao et al., 2022*). Cy5-labeled PCR products were separated on 6% native polyacrylamide gels followed by fluorescence imaging with G:BOX Chem XL (Syngene, Cambridge, UK); signals were quantitated by Image J software and exon 7 inclusion was expressed as a percentage of the total amount of spliced transcripts.

## Western blotting

Protein samples separated by 10% SDS-PAGE were electroblotted onto PVDF membranes (Millipore, Bedford, MA, USA). The blots were then probed with primary Anti-T7 mAb followed by secondary IRDye@ 680RD goat anti-mouse (LI-COR Biosciences, Lincoln, NE, USA). Anti-T7 mAb was a gift from Professor Adrian Krainer at Cold Spring Harbor Laboratory (*Sun et al., 2019*); Protein signals were detected with an Odyssey Infrared Imaging System (LI-COR Biosciences, Lincoln, NE, USA).

## Statistical analysis

Data from three independent experiments are presented as mean ± standard deviation. Statistical significance was analyzed by Student's t-test and one-way ANOVA with software SPSS 16.0 (SPSS Inc., Chicago, IL, USA).

# RESULTS

## The effects of tethered SR proteins on SMN1/2 exon 7 splicing

SR proteins with similar structures often exert different effects on splicing regulation. To better understand the underlying mechanisms, we took advantage of *SMN1/2* minigenes coupled with an MS2-tethering assay to compare and dissect the effects of four SR proteins (SRSF1, SRSF5, SRSF6 and SRSF9) on *SMN1/2* exon 7 splicing. These four

proteins are highly similar in domain structures: an N-terminal RRM1, followed by a Glycine-rich domain (GRD), RRM2, and a C-terminal RS domain (Fig. 1A). The MS2 sequence was inserted into the −36 position in intron 6 or +37 position in intron 7 of the *SMN1/2* minigenes (Fig. 1A), and the mutant minigenes were named as *MS2-In6-SMN1/2* or *MS2-In7-SMN1/2*. Regarding the choice of MS2 position, we avoided disrupting core splicing signals (splice sites, polypyrimidine regions, and branch point sequences) as well as known splicing regulatory elements, such as the intron splicing silencer at *SMN2* intron 7 positions 11–24 (ISS-N1) (*Singh et al., 2017*), and guaranteed complete components necessary for *SMN1/2* splicing. We chose intron 6 (position −36) and intron 7 (position +37) because these two positions have previously been used to investigate the regulatory effects of splicing factors on *SMN1/2* using an MS2-tethering assay (*Wu et al., 2017*; *Sun et al., 2012*). T7-tagged fusion proteins that contain MS2 coat protein (CP), and an SR protein were generated in the pCGT7 vector, named as CP-SRSF1, 5, 6, and 9, respectively. Plasmids were co-transfected into HEK293 cells and splicing was analyzed with semi-quantitative fluorescent RT-PCR.

Unexpectedly, for the three constructs (*MS2-In6-SMN1/2* or *MS2-In7-SMN2*), the percentages of exon 7 inclusion changed from 99, 35 and 5, respectively, in the presence of the empty vector, to 40, 1 and 41 with T7-CP ($P < 0.05$). The exon 7 splicing of the left construct, *MS2-In7-SMN1*, was not significantly changed with T7-CP, which can be explain by the high inclusion rate of exon 7 that concealed the splicing promotion function when CP was tethered to intron 7. Considering the fact that MS2-CP itself significantly changed exon 7 inclusion when tethered to *SMN1/2* minigene, to reveal the function of tethered SR proteins, minigenes treated with MS2-CP were used as controls in following experiment (Figs. 1 and S1).

Over-expression of the four SR proteins (without CP) on *SMN1/2* exon 7 splicing were firstly checked to distinguish MS2-dependent binding function (Fig. S1). Over-expression of the four T7-tagged SR proteins (without CP) exhibited different effects on exon 7 splicing of mutant minigenes: free SRSF1 slightly increased exon 7 inclusion in *MS2-In6-SMN2*, but had on effects on other minigenes; SRSF9 promoted exon 7 inclusion of *SMN2* but had no effects on *SMN1* (which is likely due to the gene already having a high percentage of exon 7 inclusion), while over-expression of untethered SRSF5 and SRSF6 inhibited exon 7 splicing of both minigenes (Figs. S1A–S1C). Compared to T7-CP, further attachment of the SR proteins showed complex outcomes: when tethered to *SMN1* intron 6, CP-SRSF1 markedly inhibited exon 7 inclusion, whereas CP-SRSF5 and CP-SRSF9 promoted splicing, and CP-SRSF6 had no effects (Fig. 1C). When tethered to *SMN2* intron 7, CP-SRSF1 moderately inhibited *SMN2* exon 7 splicing, whereas CP-SRSF5 and CP-SRSF6 moderately promoted inclusion of the exon, and SRSF9 failed to make any change (Fig. 1D). As *MS2-In7-SMN1* bound by T7-CP displayed extremely strong promotion of *SMN1* exon 7 splicing, no higher exon 7 inclusion can be observed, and the splicing changes made by SR proteins were diminished at this site: CP-SRSF1 can also decrease in *SMN1* exon 7 splicing. *MS2-In6-SMN2* and T7-CP system faced a similar situation: the extremely low exon 7 inclusion concealed the splicing inhibitory effects, while CP-SRSF5/SRSF9 showed limited promotion in *SMN2* exon 7 splicing (Figs. 1C and

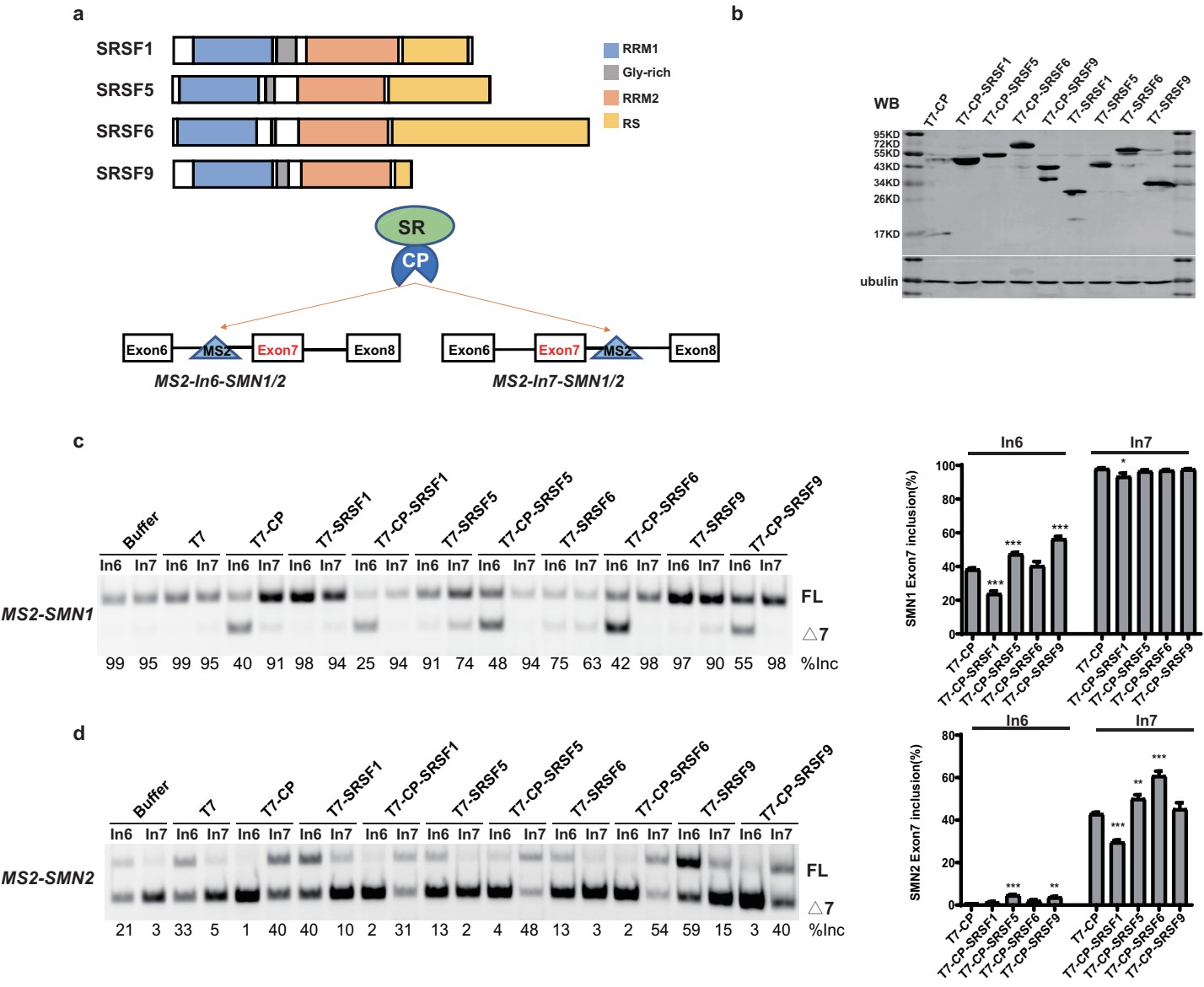

**Figure 1 SR proteins display various effects on *SMN1/2* splicing.** (A) Using MS2 coat protein (CP), the fusion SR proteins were tethered to their target motif (MS2) located in intron 6 or 7 to explore their effects on splicing at these two positions; (B) Western blot analysis of each protein expressed in HEK293 cells using an anti-T7 antibody; (C) The functional analysis of SR proteins on *MS2-SMN1* minigene; (D) The functional analysis of SR proteins on *MS2-SMN2* minigene. T7-CP is used as a control. Quantitative data from three independent experiments are shown in histograms. ***$P < 0.001$, **$P < 0.01$, *$P < 0.05$.

1D). Therefore, the CP fused SR proteins exerted significantly different function than free SR proteins did, indicating a successful tethering of SR proteins to MS2 binding site. Considering that CP-SR proteins exerted similar splicing effects in the context of *MS2-SMN1* and *MS2-SMN2* minigenes, *MS2-In6-SMN1* and *MS2-In7-SMN2* were complementarily employed for position-related analyses in this study to get rid of the defects mentioned above.

It is surprising that no two SR proteins behaved similarly in splicing regulation for both tethered sites. The data also imparts that when bound to a specific site in either upstream

or downstream intron of a cassette exon, SR proteins may inhibit or stimulate splicing, or have no effects.

## Domain analysis of the four SR proteins

To identify the essential domains that vary the functions of the four SR proteins, a series of mutants were created by deleting one or more domains: eight mutants were generated for each protein, named RRM1, RRM2, RS domain, RRM1/Gly-rich, ΔRS domain, Gly-rich/RRM2, ΔRRM1, RRM2/RS (Fig. 2A). Plasmids were co-transfected into HEK293 cells, and western blotting confirmed that all mutant proteins were properly expressed (Figs. 2B, S2, S3A–S3C).

For SRSF1 mutants, both CP-RRM1 and CP-RRM2 enhanced *SMN1* exon 7 inclusion when tethered at intron 6; all mutants that retained the RS domain suppressed splicing, suggesting that the original inhibitory effect of intron 6-tethered CP-SRSF1 was mediated by the RS domain. Combination of RRM1 and/or RRM2 with the GRD showed no effect on exon 7 inclusion, suggesting that the stimulatory role of RRM1 and RRM2 was compromised by GRD. When tethered to intron 7 of *SMN2*, almost all the mutants significantly inhibited *SMN2* exon 7 splicing except for RRM1/GRD; the effects in *SMN1* were statistically much less or not significant but we observed a similar trend (Fig. 2C).

For SRSF5 mutants when tethered at intron 6, CP-RRM1 and CP-RRM2, like their counterparts in SRSF1, stimulated exon 7 splicing. Mutants retaining the RS domain (CP-RS, CP-ΔRRM1, and CP-RRM2/RS) had no effect on exon 7 inclusion. It is likely that the RS domain neutralized the stimulatory effects of RRMs. When tethered at *SMN2* intron 7, two mutants with the RS domain in the absence of GRD (CP-RS and CP-RRM2/RS) potently inhibited *SMN2* exon 7 splicing while others moderately inhibited it or had no effect. We noticed that, in the presence of GRD, the RS domain failed to keep its inhibitory function as seen in CP-SRSF5 and CP-ΔRRM1 (Fig. 2D).

Similar to SRSF1 and SRSF5 mutants, when tethered to intron 6, SRSF6 mutants CP-RRM1 and CP-RRM2 acted as splicing activators, but GRD and the RS domain appeared to compromise their effects. When tethered to *SMN2* intron 7, SRSF6 mutants CP-RS, ΔRRM1 and CP-RRM2/RS all significantly increased exon 7 inclusion, indicating that the RS domain of SRSF6 somehow acted as a splicing activator in this scenario; CP-RRM1 inhibited *SMN2* exon 7 splicing but RRM2 appeared to have no effect (Fig. 2E).

For SRSF9 mutants, when tethered to intron 6, CP-RRM2 but not RRM1 promoted exon 7 splicing, and its effect was again impaired by GRD. When tethered to *SMN2* intron 7, CP-RRM2 inhibited exon 7 splicing, and both the RS and the GRD counteracted this effect (Fig. 2F).

The domain analysis of the four SR proteins indicates that RRM1/2 conferred splicing functions other than RNA binding, which is consistent with previous studies (*Zhang et al., 2014*; *Lee et al., 2017*); and in the context of *SMN1/2* exon 7 inclusion, they acted as splicing activators when bound to the upstream intron with only one exception of CP-RRM1 of

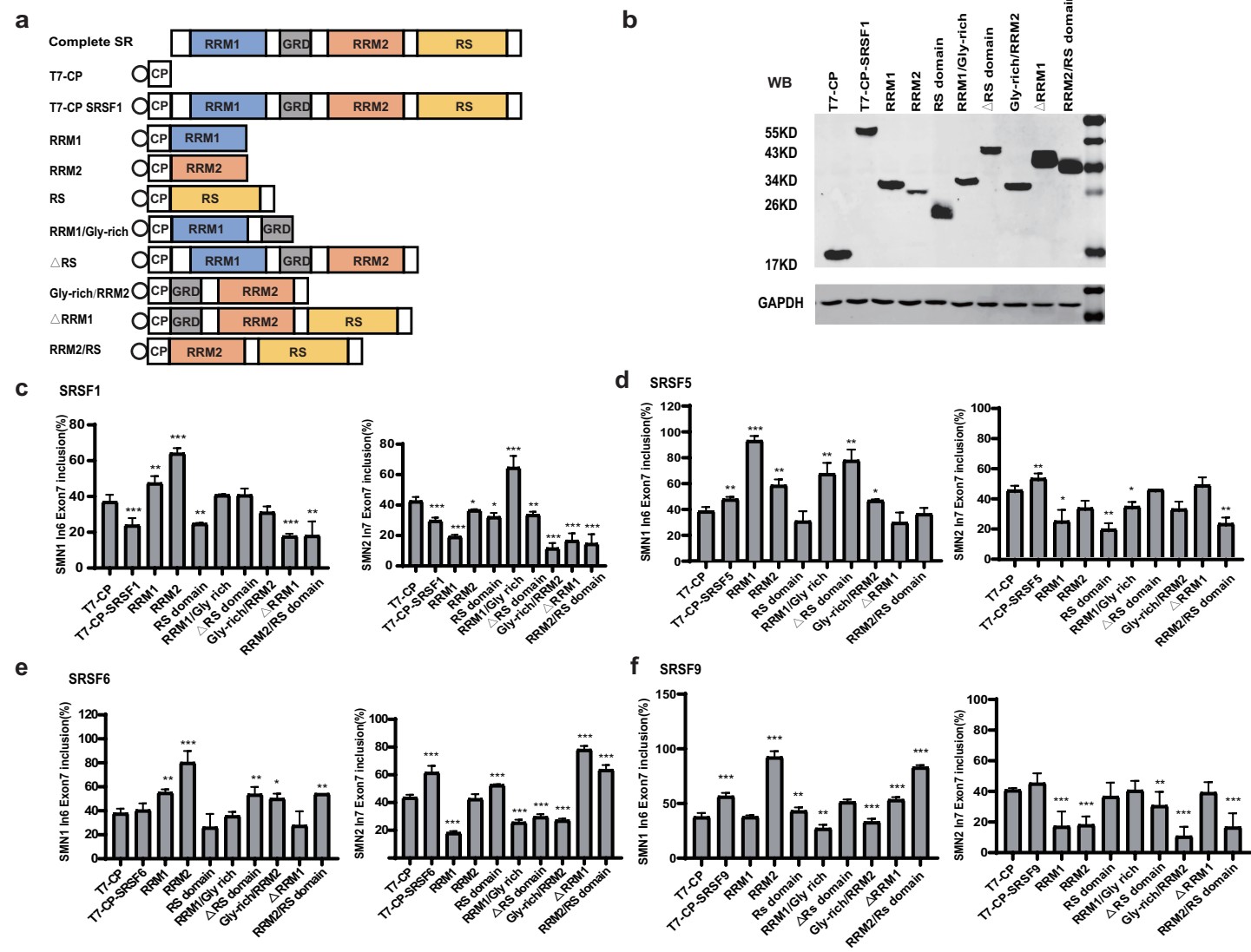

**Figure 2** Domain analysis of the four SR proteins in regulating *SMN1/2* splicing. (A) Diagram of the primary structure of the SR protein and its mutants. The T7 label is indicated by a circle; (B) western blotting (WB) was performed using an anti-T7 antibody to verify the successful protein expression; (C–F) the splicing effects of SRSF1, SRSF5, SRSF6 and SRSF9 mutants either at upstream or downstream of exon 7 in both *SMN1* and *SMN2* minigenes. T7-CP is used as a control. Quantitative data from three independent experiments are shown in histograms. ***$P < 0.001$, **$P < 0.01$, *$P < 0.05$.

SRSF9 that was neutral, whereas they acted as splicing repressors or had no effects when bound to the downstream intron.

## A conserved nonapeptide in the C-terminus of SRSF1/9 RRM1 inhibits the effect of RRM1 on splicing

When tethered at the −36 position of intron 6, RRM1 of SRSF9 is the only RRM that failed to stimulate *SMN1* exon 7 splicing, and RRM1 of SRSF1 had the least stimulatory effect compared to RRM1 of SRSF5 and SRSF6. We aligned the RRM1 amino acid sequences of SR proteins from various species (Fig. S4) and identified a conserved nonapeptide

(RLRVEFPRT) in the C-terminus of RRM1 of SRSF1 and SRSF9, hereinafter referred to as CRRM1-9P. We hypothesized that the peptide is an inhibitory motif (Figs. 3A and S4). We constructed a series of CP-fused RRM1 mutants and tested their effects on *SMN1/2* exon 7 splicing using the *MS2-In6-SMN1* and *MS2-In7-SMN2* minigene in HEK293 cells. Since the N-terminal extension of SRSF1 is an inhibitory motif (*Shaw et al., 2007*), mutants with or without this region were both investigated. As expected, deletion of CRRM1-9P significantly increased *SMN1/2* exon 7 splicing in either the absence or presence of the inhibitory N-terminus (+N) in both minigenes (Figs. 3C–3E), indicating that the nonapeptide exerts its inhibitory function at either upstream or downstream intron of the cassette exon 7.

As the charge of each amino acid may affect peptide functions, we investigated the charges of CRRM1-9P by site mutagenesis. Western blot confirmed the successful expression of each mutation (Fig. S5). When all amino acids were mutated to uncharged Ala, or negatively charged Glu, exon 7 inclusion was strongly increased and reached an extremely high level (Fig. 3F), indicating a total functional loss of the mutants. In contrast, when all amino acids were mutated to positively charged Arg, the lowest increment of exon 7 inclusion were observed, suggesting Arg residues may contribute to the inhibitory nature of CRRM1-9P. To test this hypothesis, we generated three mutants that changed one, two or three Arg residues into Glu residues, as CRRM1-9P contains three Arg residues. The more Arg residues were switched to Glu residues, the higher exon 7 inclusion percentage was observed, confirming the key role of Arg residues for the inhibitory nature of the nonapeptide (Fig. 3F).

## A regulatory motif was identified at the N-terminus of RRM1 in both SRSF5 and SRSF6

RRM1 of SRSF5 potently and RRM1 of SRSF6 moderately promoted *SMN1* exon 7 splicing when tethered to intron 6 (Figs. 2D and 2E). To further investigate the effects, we split SRSF5/SRSF6 RRM1 into four parts with equal length, named A, B, C and D. Each fragment contains 18 (A, B and C) or 17 (D) amino acids (Fig. 4A). CP-fused RRM1 mutants of both SR proteins harbor one part or a combination of multiple parts (Figs. 4B and S3D). When tethered to intron 6, fragment A of both proteins displayed the strongest stimulatory effect on *SMN1* exon 7 splicing, followed by fragments B and C. Fragment D of SRSF6 weakly increased exon 7 inclusion, while fragment D of SRSF5 had no effect (Figs. 4C and 4D). The data suggests that the stimulatory activity of RRM1 lies in the N-terminus of the domain.

To further map the key motif in fragment A that promotes exon 7 splicing in *MS2-In6-SMN1* minigene, a series of deletion mutations were generated from both the N-terminal and C-terminal ends of the fragment. For SRSF5 fragment A, deletion of the N-terminal two or more amino acids caused marked reduction in its activity, and a significant activity reduction was also observed when over ten amino acids were deleted from the C-terminus (Fig. 5A). For SRSF6 fragment A, deletion of the first four or more amino acids, or the last eight amino acids markedly impaired its activity (Fig. 5B). The deletion analysis indicates

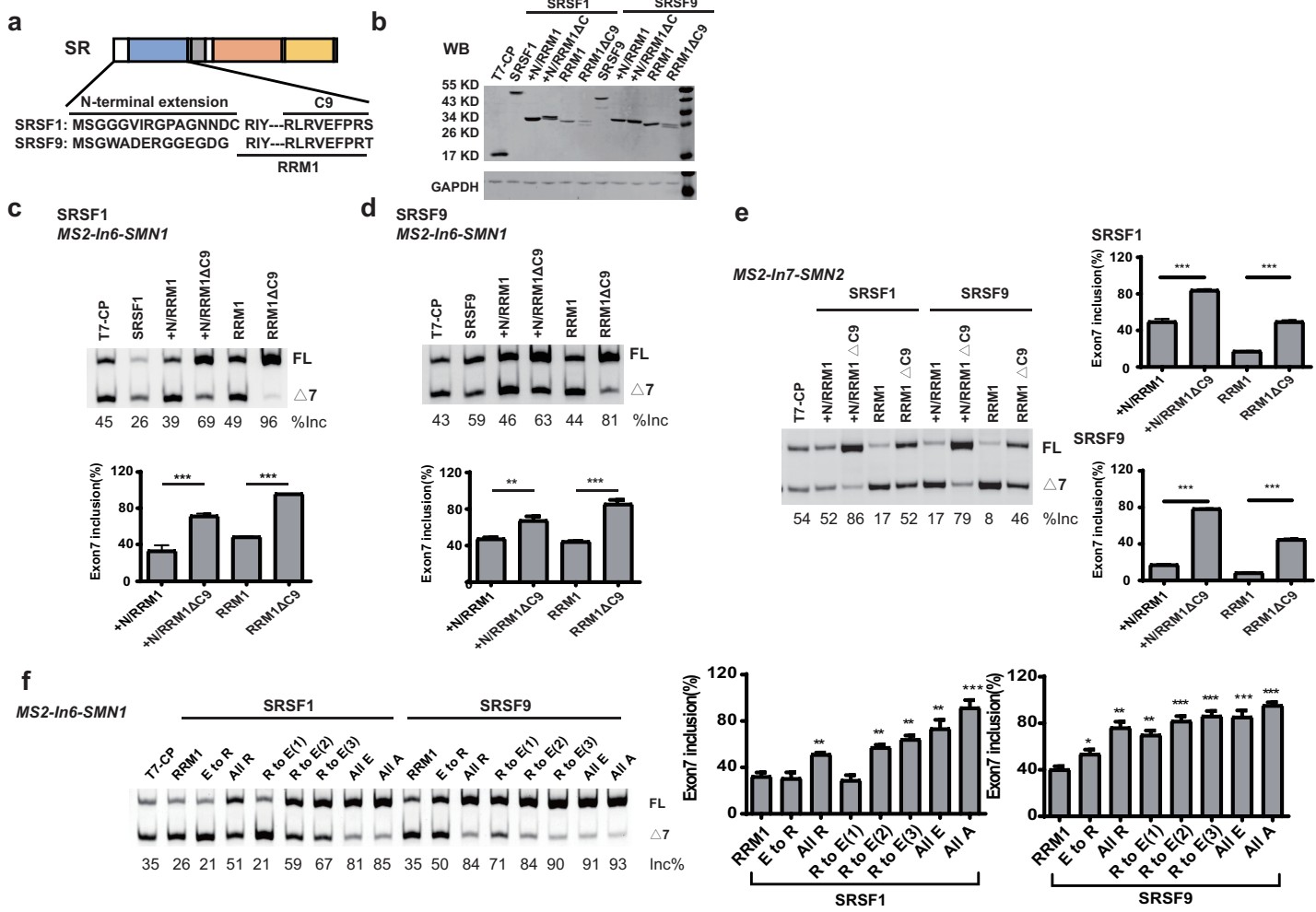

**Figure 3 The detection of an inhibitory nonapeptide at the C-terminus of RRM1 in SRSF1/9.** (A) Diagram depicting amino acids in the N-terminus extension and RRM1 C-terminus of SRSF1/9; (B) the successful expression of SRSF1/9 RRM1 truncated proteins was verified by western blotting (WB); (C and D) the splicing effects of SRSF1/9 RRM1 mutants in *MS2-In6-SMN1*; (E) the splicing effects the nonapeptide in *MS2-In7-SMN2*; (F) the Arginine residue play key roles in the regulatory function of the nonapeptide. C9, the conserved nonapeptide at the C-terminus of SRSF1/9 RRM1; +N, the N-terminus extension of RRM1. Quantitative data from three independent experiments are shown in histograms. ***$P < 0.001$, **$P < 0.01$, *$P < 0.05$.

that the first eight amino acids (CRVFIGRL) in SRSF5 RRM1 and 10 amino acids from the 3rd to 12th (RVYIGRLSYN) in SRSF6 RRM1 are the minimal functional motifs.

Alignment of the two sequences identified a highly conserved sequence of seven amino acids, RVFIGRL for SRSF5, in which only one amino acid Phe is replaced with a structurally similar Tyr in SRSF6, hereinafter referred to as NRRM1-7P (Fig. S6). Considering that the deletion of the first two amino acids of SRSF5 RMM1 caused a significant decrease in *SMN1* exon 7 splicing, and same deletion in SRSF6 RRM1 also slightly inhibited splicing (Figs. 5A and 5B), we hypothesized that these deleted amino acids could increase the function of NRRM1-7P. Therefore, four peptides, NRRM1-7P of SRSF5 and SRSF6 with or without their proceeding amino acids were further tested. The heptapeptide motif itself pronouncedly promoted exon 7 splicing when tethered to

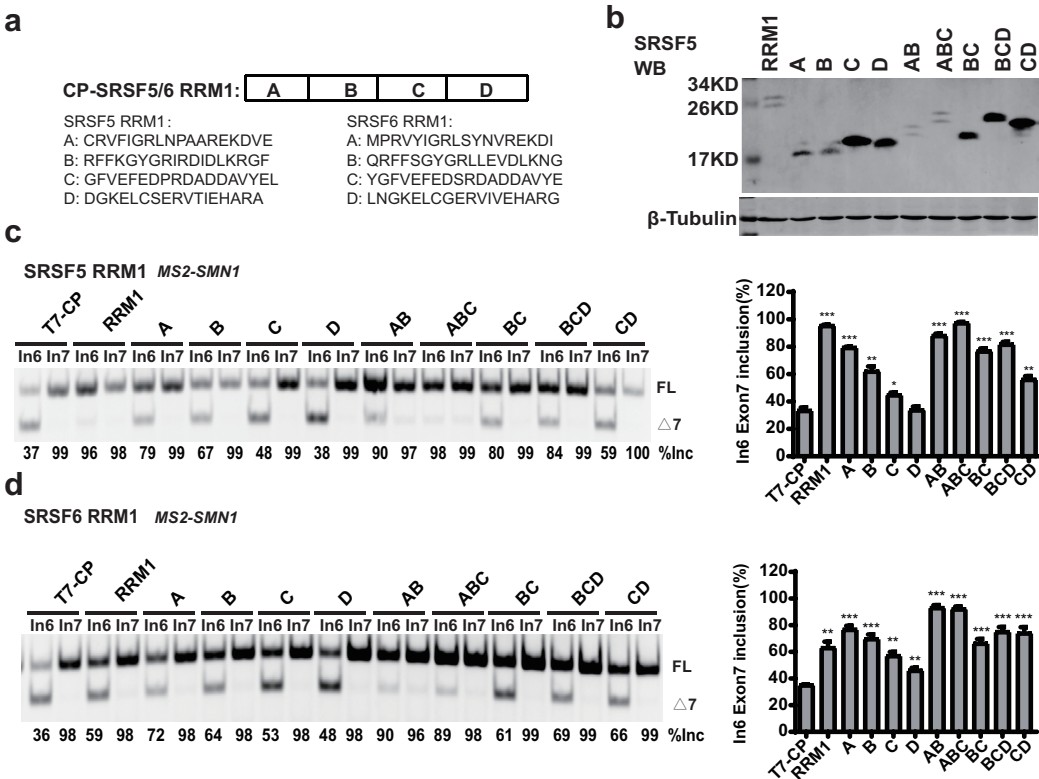

**Figure 4 The identification of key structures in SRSF5/6 RRM1 affecting splicing.** (A) Diagram of the primary structure of SRSF5/6 RRM1. RRM1 is roughly divided into four fragments; (B) the successful expression of the truncated proteins was verified by western blotting (WB); (C and D) the splicing effect of SRSF5/6 RRM1 mutants binding to the upstream of exon 7 in *SMN1*. T7-CP is used as a control. Quantitative data from three independent experiments are shown in histograms. ***$P < 0.001$, **$P < 0.01$, *$P < 0.05$.

intron 6; however, the amino acid Cys proceeding NRRM1-7P was critical for the potent stimulatory effect of fragment A derived from SRSF5, this peptide hereinafter referred to as NRRM1-8P (Fig. 5C). We further investigated the splicing effects of the heptapeptide on *MS2-In7-SMN2*. Interestingly, contrary to the results of *MS2-In6-SMN1*, all the four CP-fused peptide strongly inhibited *SMN2* exon 7 splicing (Fig. 5C), indicating that the regulatory motif of SRSF5/SRSF6 RRM1 is position-dependent. Similar to CRRM1-9P, NRRM1-7P also contains two Arg residues. We wondered whether Arg residues are also crucial to its regulatory function by mutating one or both Arg residues to either Glu or Ala. As expected, the more Arg residues were mutated, the lower level of exon 7 inclusion was observed (Fig. S7).

## NRRM1-8P affect splicing without MS2 tether in various gene settings

Using the MS2 tethering splicing assay, we attempted to validate the functional short peptide at other genes with classical splicing regulation. We chose the *MAPT* and *GOLM2* genes to insert MS2 in the upstream and downstream introns of exon 10/9, referred to as *MS2-In9/In10-MAPT* and *MS2-In8/In9-GOLM2*. After transfection, we discovered that the short peptide exerted a repressive function at all positions in both genes, except *MS2-*

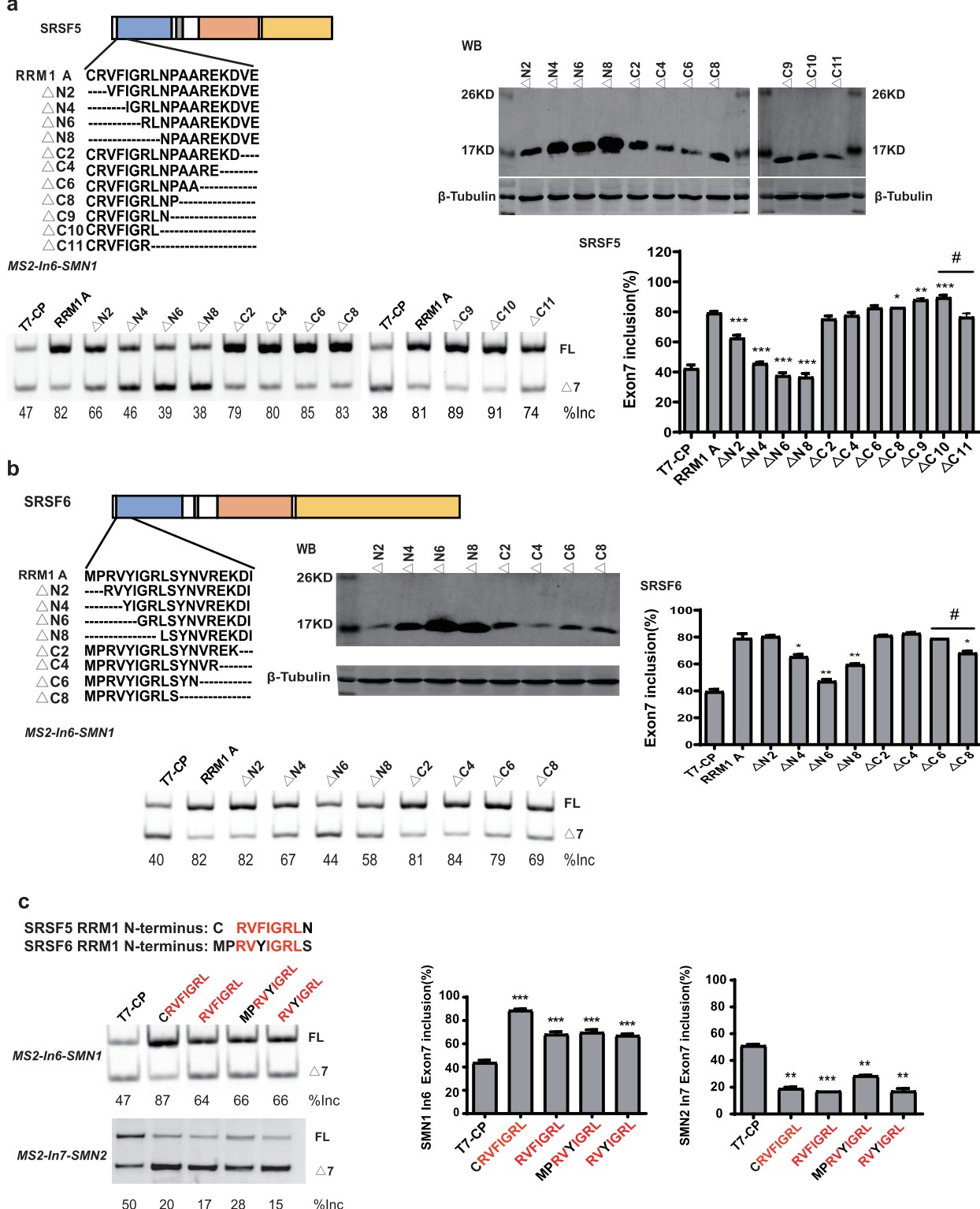

**Figure 5  A regulatory motif was identified at the N-terminus of RRM1 in both SRSF5 and SRSF6.** (A) The analysis of the effects of amino acids deletion in RRM1 fragment A in SRSF5, RRM1 A is used as a control. Quantitative data from three independent experiments are shown in histograms. $P < 0.001$, $P < 0.05$ *vs.* RRM1A; [#]$P < 0.05$ *vs.* ΔC10; (B) the analysis of the effects of amino acids deletion in RRM1 fragment A in SRSF6,

**Figure 5 (continued)**
RRM1 A is used as a control. Quantitative data from three independent experiments are shown in histograms. $P < 0.001$ *vs.* RRM1A; $^{\#}P < 0.05$ *vs.* ΔC6; (C) a conserved heptapeptide in SRSF5/6 RRM1 N-terminus regulate exon 7 splicing in *MS2-In6-SMN1* and *MS2-In7-SMN2*. T7-CP is used as a control. Quantitative data from three independent experiments are shown in histograms. $P < 0.001$, $P < 0.01$, $P < 0.05$ *vs.* T7-CP.

*In9-GOLM2*, where the inclusion percentage was low and the repressive effect is not insignificant. Those results demonstrated that the background position sequences might interfere with the activity of the short peptide (Fig. S8).

To rule out the effect of MS2-tethering, we next asked whether the short peptide from SRSF5/6 can regulate splicing when not tethered in the minigenes. Plasmids expressing T7-CP-fused NRRM1-8P and the WT *SMN1* or *SMN2* minigene were co-transfected into HEK293 cells, and *SMN2* exon 7 splicing was analyzed. As expected, the peptide caused a robust decrease in *SMN2* exon 7 inclusion despite that the effect on *SMN1* was much weaker (Figs. 6A and 6B). The RNA binding region of CP is 32–105 amino acids, according to the study (*Rolfsson et al., 2016*). We then deleted this region of the CP further to study the regulatory effect of the short peptide regulatory on the WT *SMN2* minigene. Despite the lack of RNA-binding ability of CP, our short peptide was able to decrease the splicing of WT *SMN2* (Fig. S9).

We further tested whether the inhibitory effect of the peptide can be observed in other gene contexts. We took advantage of two previously-described *microtubule associated protein tau* (*MAPT*) and *Caspase 3* (*CASP3*) minigenes, both harboring an alternatively splicing cassette exon (*Gao et al., 2022*). Similar to what we observed in the *SMN2* minigene, overexpression of CRVFIGL significantly repressed splicing of the cassette exon for both minigenes (Figs. 6C and 6D). These data suggest that NRRM1-8P can be broadly exploited to regulate different alternative splicing events.

## DISCUSSION

The SR proteins are involved in many diseases and can be even considered as key oncogenes (*Kedzierska & Piekielko-Witkowska, 2017*). Their role in regulating spinal muscular atrophy (SMA) has long been discussed. In this study, four SR proteins were thoroughly investigated; none of them had similar effects on *SMN1/2* splicing, and the binding site had a strong influence on the activities of SRSF5 and SRSF9. The key functional domains of each SR protein were then characterized, especially that RRMs were observed to exert an RS-domain-independent effect on splicing. Further investigations focused on the conserved RRM1 to reveal the key structures of each SR protein resulted in the detection of an inhibitory nonapeptide in the C-terminus of SRSF1/9 and a short peptide in the N-terminus of SRSF5 or SRSF6, which is a strong splicing enhancer. If managed properly, these short peptides can serve as splice-switching molecules.

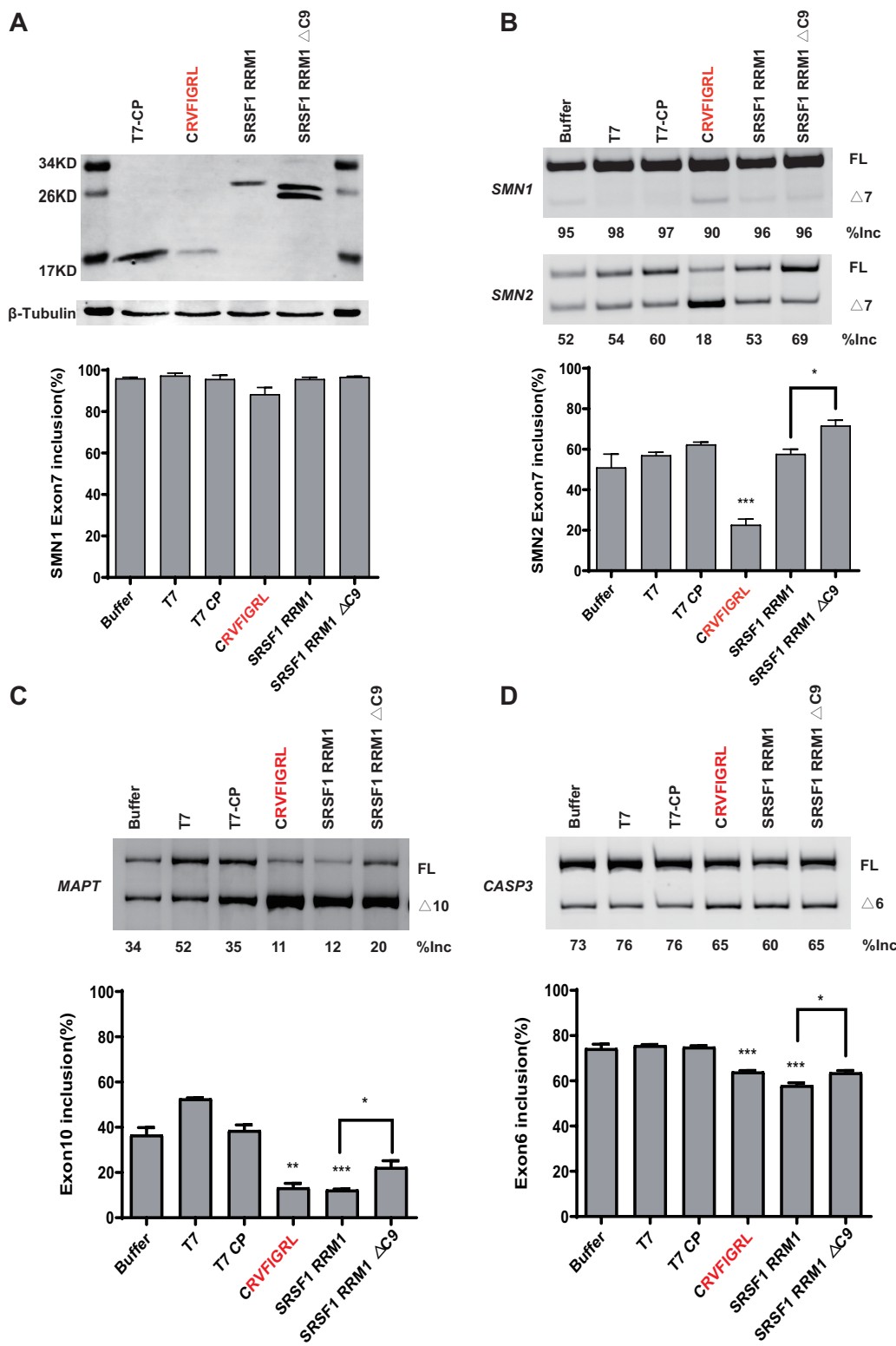

**Figure 6 The short peptide motif displayed similar splicing effects on *MAPT* and *CASP3* minigenes.**
(A) The confirmation of the successful expressions of the proteins by western blotting. (B) The splicing effects of each overexpressed proteins on wild type *SMN1/2* minigenes. (C) The splicing effects of each

**Figure 6** (continued)
overexpressed proteins on *MAPT* minigene; (D) The splicing effects of each overexpressed proteins on *CASP3* minigene. T7-CP is used as a control. Quantitative data from three independent experiments are shown in histograms. ***$P < 0.001$, **$P < 0.01$, *$P < 0.05$.

## MS2-tethering assay confirmed that the effect of SRSF6/9 on splicing was position-dependent

The MS2-tethering system are frequently used to investigate the regulatory functions of splicing factors at different positions, but MS2-CP has been observed to affect splicing may be due to steric hindrance. *Shen & Mattox (2012)* studied the effect of splicing on Tra2 tethering to an intron by fusion with the MS2 coat protein. *Sun et al. (2012)* used an MS2-tethering test to investigate the mechanisms of RBFOX splicing activation and repression. *Qiu et al. (2022)* used an MS2-tethering array and uncovered that the effects of hnRNP A1/2 on *SMN1/2* exon 7 splicing are position-dependent. In these studies, CP were all found to somehow changed the splicing of the target gene. Our team recently revealed that the MS2-tethering method could investigate the splicing effect of HuR when it binds to *SMN2* intron 6, and the CP hindrance exerts little effects when analyzing the interactions between HuR and other splicing factors (*Wu et al., 2017*). Admittedly, the MS2-CP method would introduce variations, but it is still believed to be an optimal choice for functional analysis of splicing factors. Therefore, the same MS2-tethering system, including *MS2-In6/7-SMN1/2* minigenes, was utilized in the current study to analyze SR protein functions. The empty vector with CP was used as the control to exclude the splicing effect caused by the CP backbone. We need to get rid of CP for the next in-depth mechanism analyses.

   To rule out the influence of the binding site of SR proteins, we additionally created a new minigene named *MS2-In7-SMN2**. The MS2 binding sequence was inserted in *SMN2* intron 7 at positions 11 to 15, which does not affect the binding site of SRSF1. *MS2-In7-SMN2** was used to compare with *MS2-In7-SMN2*, which had an SRSF1 binding site disruption. T7-SRSF1 strongly inhibited exon 7 splicing in *MS2-In7-SMN2**, which proved a good maintenance of the SRSF1 binding site. The inhibitory effect is in consistence with CP-SRSF1 on *MS2-In7-SMN2*, proving that the MS2-tetherd SRSF1 has been successfully driven to its original binding site. What's more, in the context of *MS2-In7-SMN2**, the T7-CP construct (150 ng) and T7-SRSF1 (150 ng each) were co-transfected with the *MS2-In7-SMN2** minigene, co-expression of two separate proteins resulted in a different effect on exon 7 splicing, compared to T7-CP-SRSF1. We confirm that the repression of exon 7 splicing by SR fusion protein is because they are fused to MS2 CP (Fig. S10). Taken together we believe that the T7-CP-SR fusion proteins will preferentially bind to the MS2 position, which can eliminate the influence of the SR protein when both SR and CP binding sites are present in the pre-mRNA. Our results based on this MS2-tethering method is reliable.

   Recent studies have revealed that RBPs may achieve their functions *via* binding to their targets in a position-dependent pattern on RNAs. Global sequencing studies have revealed

that multiple RBPs affect pre-mRNA splicing in a position specific manner. For example, CLIP-Seq data revealed Mbnl could repress splicing when bound to the upstream/inside intron region of the alternative exon (*Wang et al., 2012*); the functional sites of ESRP1 were enriched at the downstream of ESRP-enhanced exons (*Dittmar et al., 2012*). Some RBPs even display opposite functions at different positions. Rbfox1/2 and CELF2 have been shown to be suppressors when bound to the upstream of the target exons, but activators when bound to the downstream region (*Sun et al., 2012*; *Ajith et al., 2016*). As shown in our results, the position-independent splicing functions of SRSF6/9 were confirmed. Traditionally, when SR proteins bind to intronic sequences, they are thought to promote exon skipping (*Zhou & Fu, 2013*), but their interactions with the exonic target are intricate (*Han et al., 2011*). SRSF6 was indicated to suppress splicing when bind to the downstream 5′ splice site (*Erkelenz et al., 2013*), and SRSF9 was found to function as a repressor of 3′ splice site utilization (*Simard & Chabot, 2002*). Surprisingly, our data recorded the opposite splicing effects of SRSF6/9 on splicing. SR proteins commonly regulate splicing through the interaction with other co-factors (*Cho et al., 2011*; *Tripathi et al., 2012*). Since we used different minigene systems, which might be regulated by different co-factors causing the opposing effects. Therefore, the mechanism behind SR proteins regulation is more complicated than previously thought, our findings will contribute to a better understanding of pre-mRNA splicing regulation.

## The RRM(s) display the RS-domain-independent splicing

The RS domains of SR proteins were thought to be essential for constitutive splicing, as evidence showed that SR protein lacking its RS domain was unable to complement S100 for splicing of constitutive substrates (*Shaw et al., 2007*). Subsequently, researchers have indicated that the RS domain is actually dispensable for splicing in some cases (*Shaw et al., 2007*). The requirement for the RS domain is substrate specific, and correlates with the strength of the splicing signals (*Zhu & Krainer, 2000*). Here in our study, the essential role of the RS domain in regulating splicing was validated using a number of mutation analysis, and it can either activate or suppress splicing through different mechanisms (*Shaw et al., 2007*). Interestingly, in the RS-domain-dependent manner, the rest of the protein structures, especially RRM1/2 showed diversified splicing effects, supporting the notion that the RS domain is dispensable for some substrates. A previous study of SRSF1 pointed out the potential mechanism, the RRM1 specifically interact with the protein phosphatase 1 (PP1) to affect *SMN2* splicing (*Aubol et al., 2018*), as the inhibition of PP1 has been proved to enhance *SMN2* exon 7 inclusion in the mouse model (*Novoyatleva et al., 2008*).

Our data appears to be contrary to the notion that SR proteins mainly function as intronic repressors, and one possible explanation is that the breakdown of the direct interaction between the RS domain and pre-mRNA renders an SR protein ineffective. As the RS domain was previously indicated to directly interact with the branch point (BP) during E-complex and connect to the 5′ splice site during B complex to inhibit cassette exon splicing (*Hertel & Graveley, 2005*; *Moon et al., 2017*). Either the deletion of the RS domain or the mis-location caused by the MS2 tethering assay would prevent this interaction. The loss of the protein-protein interaction between the RS domain and other

splicing factors like U2AF might be the other reason resulting in totally different splicing effects in the RS-domain-independent regulation (*Wang, Xiao & Manley, 1998*). It is worth mentioning that the GRD is observed to restrain the function of RRM1/2. A study on engineering splicing factors (ESFs) reported a similar pattern of repression of cassette exon inclusion by GRD (*Mao et al., 2018*). As the Glycine-rich interdomain linker allows the binding of the RRM1 to RNA either upstream or downstream of RRM2-binding site at two precise positions (*Clery et al., 2021*), its significant impact on RRMs is reasonable. Therefore, the RRMs of SR proteins can exert distinct regulatory functions in an RS-domain-independent splicing, though highly affected by other domains.

## CRRM1-9P in SRSF1 and SRSF9 and NRRM1-7P derived from SRSF5 or SRSF6 can act as potent splicing modulators

RRM1 can have N-terminal and C-terminal extensions augmenting their core structures that are usually poorly ordered, but in some cases adopt a secondary structure (*Maris, Dominguez & Allain, 2005*). As RRM1 of SRSF1 or SRSF9 weakly affected splicing at the upstream of exon 7, it was assumed that they might harbor some splicing inhibitory elements. Here we report for the first time a highly conserved nonapeptide (RLRVEFPRT) in the C-terminus of RRM1 of both SRSF1 and SRSF9 functioning as a strong splicing suppressor in the absence of RS-domain. Previously, the N-terminus of SRSF1 protein has been thoroughly explored and was revealed to contribute to the inhibitory effect of the RRM1 extension on splicing (*Clery et al., 2021*; *Shaw et al., 2007*). The deletion of the N-terminus extension of SRSF1 can slightly increase exon 7 inclusion (Fig. 3C); however, the deletion of the N-terminal extension of SRSF9 made no difference on *SMN1* exon 7 splicing (Fig. 3D), implying the existence of other inhibitory structures in RRM1. A nonapeptide motif in the C-terminus of SRSF1 and SRSF9 RRM1 was found to exert a much stronger splicing inhibition, indicating that the key inhibitory structure affecting RRM1 is located at the C-terminus of RRM1 instead of its N-terminal extension.

The splicing regulator PP1 was reported to interact with the RVDF motif in Tra2-β1 (*Handa et al., 1999*), while the SRSF1 and SRSF9 were also found to bind to PP1 by employing the conserved RVXF motif in the β4-binding region (*Phelan et al., 2012*). Therefore, the recruiting of PP1 through the RVXF motif may be the key mechanism of the nonapeptide in regulating *SMN1/2* splicing. Other possibilities for explaining the function of C-terminal nonapeptide is that this peptide region forms beta-strand in RRM1, the deletion or mutation of this peptide will damage structural intactness of RRM1. But considering that the C-terminus deletion of SRSF5/6 exerted little effecting on RRM1 function, further functional studies in regard to β4 region is needed.

Furthermore, our data indicated that the extreme N-terminus of SRSF5 or SRSF6 RRM1 acted as a strong splicing regulator (Fig. 5). Their core functional structure was found to be a highly conserved heptapeptide-RVY(F)IGRL, with a molecular weight ranging from 0.86–0.88 kDa, located in the β1-folding region. The N-terminus of RRM1 is believed to be flexible and can adopt many conformations to modulate splicing either positively or negatively (*Clery et al., 2021*). Our data showed that the heptapeptide can modulate splicing of wildtype *SMN2* minigene, implying that this peptide motif may retain the RNA

binding ability to directly target *SMN2* minigene, which is consistent with the previous finding that the RRM binds to RNA by the motif of β-folding region (*Allain et al., 2000*).

Another possible explanation causing the regulatory effects of these short peptides are due to the presence of charged amino acids. It has been reported that the splicing regulation is tightly interconnected with the physicochemical properties of splicing factors, and domains harboring polar charged amino acids promote exon inclusion while domains with uncharged amino acids repress exon inclusion (*Mao et al., 2018*). In a recent study, it has been demonstrated that the mutation in SF3B1 causes splicing errors are due to the substitution of the location and charges of amino acids (*Canbezdi et al., 2021*). Our follow-up experiment validated this hypothesis, when the charged amino acids were mutated the loss of function of this nonapeptide was observed.

### The *NRRM1-7P* can be utilized as potential therapeutic tool

Peptides are versatile and attractive biomolecules that can be applied to modulate genetic mechanisms like alternative splicing. Small peptides, ideally less than 1 kDa have long been studied as efficient therapeutic drugs that can overcome many delivery issues (*Berdnikova et al., 2021*). Besides their application as carriers, natural and synthetic peptides are resources for providing molecules that may interact with the spliceosome and/or alter molecular pathways (*Nancy, Nora & Rebeca, 2015*). In a previous study, peptide inhibitors of pre-mRNA splicing were derived from the splicing factors CDC5L and PLRG1 at the sites surrounding phylogenetic highly conserved amino acids (*Ajuh & Lamond, 2003*). Here, the heptapeptide named NRRM1-7P was identified from two other important splicing factors SRSF5 and SRSF6, and NRRM1-7P of SRSF5 with its proceeding amino acid Cys was found to exists the strongest effects on *SMN1/2* splicing. Due to its modest size, NRRM1-7P have the strong potential to accommodate additional elements to exerts even higher therapeutic effects. One choice is using cell penetrating peptide (CPP), which has been proved to be effective in oligonucleotide that can help regulate a single splicing event and restore correct gene expression (*Shiraishi & Nielsen, 2011*). The other choice is employing specific RNA-binding elements, antisense oligonucleotide (ASO) with a binding site of TDP-43 protein have shown profound effects on target-specific splicing (*Brosseau et al., 2014*). In addition, NRRM1-7P can be used as bifunctional component to improve other gene therapies, like ASOs. Researchers have been developing ASOs with enhanced functions for years and made a great progress. For example, an ASO designed on *SMN2* exon 7 with a tail of splicing enhancer GGA repeat displayed a bifunctional promotion in splicing regulation (*Skordis et al., 2003*). Therefore, the short peptide derived from SRSF5 may also be considered as a new tool for developing a new type of bifunctionally splicing switching ASOs or other molecules.

### CONCLUSIONS

This study brings a new insight into the wide field of alternative splicing regulation by SR proteins. Our data indicates that peptides are useful tools for exploring the splicing mechanism and may facilitate the future development of splicing regulators. We will further optimize and synthesize the peptides to analyze the splicing regulation ability.

Though finding efficient substances that possess specificity for a certain step of splicing is still challenging, improvements to the peptide motifs reported here will open up new avenues for producing therapeutic molecules and benefit many patients.

### Funding
This work was supported by the National Natural Science Foundation of China (grants 81271423, 81471298 and 81530035) and the, China Postdoctoral Science Foundation (grants 2021M691629). The funders had no role in study design, data collection and analysis, decision to publish, or preparation of the manuscript.

### Grant Disclosures
The following grant information was disclosed by the authors:
National Natural Science Foundation of China: 81271423, 81471298 and 81530035.
China Postdoctoral Science Foundation: 2021M691629.

### Competing Interests
The authors declare that they have no competing interests.

### Author Contributions
- Tao Jiang conceived and designed the experiments, performed the experiments, prepared figures and/or tables, and approved the final draft.
- Li Wang conceived and designed the experiments, prepared figures and/or tables, and approved the final draft.
- Liang Tang analyzed the data, authored or reviewed drafts of the article, and approved the final draft.
- Azhar Zeb analyzed the data, authored or reviewed drafts of the article, and approved the final draft.
- Yanjun Hou performed the experiments, authored or reviewed drafts of the article, and approved the final draft.

### Data Availability
The raw gels are available in the Supplemental Files.

### Supplemental Information
Supplemental information for this article can be found online at http://dx.doi.org/10.7717/peerj.16103#supplemental-information.

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
