# Peer review of "Identification of two short peptide motifs from serine/arginine-rich protein ribonucleic acid recognition motif-1 domain acting as splicing regulators"

_PeerJ, doi:10.7717/peerj.16103_

## Round 0.1 · original submission · Major Revisions

All reviewers have important suggestions, and I find reviewer #1's observations especially critical. Please respond to them all thoroughly.

Other issues:
line 107 "was inserted into the +37 position of the SMN1/2 exon 7" I think authors mean "was inserted into the +37 position of the SMN1/2 INTRON 7"

Your analysis of the effects of SRSFs tethering on the splicing relies only on comparisons between T7-CP-SRSF and T7-CP. However, Figure S1 (which should probably be part of the main text) shows that there are effects in the absence of MS2-CP interaction. You should (as reviewer #1 states) ensure that your analysis is not confounded by the effects of untethered SRSFs shown in FIgure S1. This is a critical point

Reviewer 1 ·

Basic reporting

.

Experimental design

.

Validity of the findings

.

Additional comments

[The reviewer has already reviewed this ms for another journal and recommended rejection on the grounds that the evidence was weak. A copy of the review is below.]


In this manuscript, Jiang and colleagues analyse the effects in splicing of SMN2 exon 7 when SR proteins, or portions thereof, are tethered via the MS2 system to single sites in introns 6 and 7. The purpose of the research was to probe the reasons why superficially similar proteins might exert different effects on splicing. They conclude that different SR proteins do indeed exert different effects and that these can be recapitulated by either the RNA-binding domains alone or even by short sequences from these domains.

It is not clear to me how we should interpret this work, for several reasons.

1. The effects on the MS2 system of fusing SR proteins to the MS2 coat protein (CP) are over-shadowed by the system itself. Just adding the MS2 CP (fused to a T7 tag, as in all constructs) when there is a binding site in intron 6 causes a majority of the otherwise completely included exon 7 in SMN1 to be skipped; similarly, it causes substantial inclusion of the otherwise skipped exon 7 of SMN2 when the tethering site is in intron 7 (Fig. 1). No explanation is given for this. It is conceivable that the MS2 fusion protein disrupts the binding site for other proteins or that it interferes with RNA secondary structure by stabilizing formation of the MS2 hairpin. This is a very unsatisfactory starting point, as it implies that the effects of the SR proteins might be simply to modulate the existing activity of the MS2 CP and that there are no physiologically relevant implications.

2. SRSF 1 and 9 proteins that are not fused to the MS2 CP have strong effects on the splicing of SMN2 exon 7 that depend on the location of the tethering site, even though they would not be expected to bind to the site. In both cases, inclusion is stimulated strongly when the site is in intron 6, but not when it is in intron 7. Again, one wonders what is going on and how it relates to splicing. Strikingly, these effects are greater than those seen with the SRSF-MS2 CP fusions proteins (Fig 1d).

3. The assay is fundamentally limited by the observation that the intron 7 insertion in SMN1 produces saturating levels of inclusion when T7-CP is added, and there appears to be little effect of SR proteins, and the intron 6 insertion in SMN2 produces complete skipping when T7-CP is added (Fig. 1). This restricts the assays in all subsequent figures to effects on SMN1 with the intron 6 site and SMN2 with the intron 7 site. It is not really possible to talk (as the authors do on lines 22-23 in the abstract and lines 301-302 in the Discussion, as examples) about opposite effects on different sides of the exon, as they are actually assaying different genes.

4. The use of fragments of folded domains requires some justification. While there are effects of short peptide sequences, can they be of any relevance to the mechanism of action of the intact proteins in normal circumstances?

5. There are no mechanistic investigations. For example, splicing assays might have been done in vitro to test whether the effects are actually the result of effects on transcription rates.

6. The title is therefore invalid. There is no evidence that these motifs are critical in the regulation of splicing per se or that they act in physiologically relevant circumstances

Minor points:
Lines 351-2 refer to reference 55 with regard to the effects of classes of amino acids in regulatory factors. Reference 55 actually describes the connection between the binding preferences of such factors and the protein sequence encoded by the exon being recognised. A better reference would be Mao et al. (2018), Cell Systems 7, 510-520, although the results in that case refer to intrinsically disordered regions rather than folded and stable domain structures.

Reviewer 2 ·

Basic reporting

In this manuscript, authors took a series of mutation analyses to investigate the splicing effects of four SR proteins tethered to SMN1/ SMN2 minigenes. This study spared great efforts in digging the new viewpoint of SR proteins that regulates pre-mRNA alternative splicing, and with massive of data, they are managed to identify two splicing-regulatory peptide motifs, among which a stimulatory heptapeptide at SRSF5/6 N-terminus showed strong potential for broad application, which is quite interesting. The manuscript is well-written in general, the results are clear, well explained by the figures, well commented, and interpreted. I recommend it for publication after addressing the following issues:

1. To better follow the splicing assays, it would be helpful to describe differences in exon 7 splicing in SMN1 and SMN2 mRNAs. What is the rationale and expectation for using both SMN1 and SMN2?
2. It is suggested that throughout the manuscript particular splicing reporters are referred to in the text, not just “exon7 splicing”.
3. In discussion (lines 306-315), the authors should mention recent advances on position-dependent effects of RBPs. For example, following reference could be useful.
Position-dependent effects of hnRNP A1/A2 in SMN1/2 exon7 splicing. PMID: 36208849
4. A few typos and grammatical errors are found. (i.e. Line16. “Serine/arginine-rich (SR) proteins regulates pre-mRNA splicing.”).
5. In line 35 (introduction), polypyrimidine tract and branchpoint site belong to the core splicing signals and are not secondary splicing signals as mentioned by the authors.
6. At the beginning of line 40, the two spaces are redundant.
7. Line 53, should be "spliceosome assembly" not "splice assembly".
8. Line 99 is missing a period.
9. In Materials & Methods, the authors should describe in detail how the fusion proteins and mutants are produced.
10. Figure 1b, T7-CP-SRSF9 expresses two major products, which one is the correct product?
11. Figure 1d, see lane 7 (CP-SRSF6 on In6), the labeled inclusion ratio does not match the actual one, and the authors should carefully check the original data, explain the reasons and make corrections.
12. The results should be described in order, for example, figure s4 should preferably not appear before figure s3 e&f, and figure s6 should not appear before figure s5.
13. The legends of the supplementary figures (s5 and s7) lack statistical descriptions.
14. See the figure legends, p-value “=” or “<”?

Experimental design

No comment

Validity of the findings

No comment

Reviewer 3 ·

Basic reporting

This manuscript uses MS2 tethering to study how SR proteins with tandem RRMs regulate alternative splicing of SMN1/2, and examined how these regulatory roles depend on the location of SR proteins. Overall, this study reported new information about SR proteins. The manuscript is well written, and essential references are provided, and raw data are trustable.

Experimental design

the method session and figure legends don’t provide enough information, as detailed below.
1. The detailed DNA sequences of SMN1/2 gene with MS2 inserted should be provided.
2. What antibody is used for Fig. 1b?
3. Fig 2, Gly-rich is used. while in the text GRD is used. Use one name.
4. Fig 1c and d should have a schematic panel to show which band is exon7 inclusion, which one is not.

Validity of the findings

One problem is the potential artifacts caused by the method. SR proteins can binds to exonic or intronic RNA elements, and then recruit other splicing factors. This study mainly insert MS2-binding RNA sequences in introns. The authors should select a control system to confirm the MS2 tethering method will yield similar results for well known model genes, such as IgM or beta-globin. MS2-binding sequences should also be inserted in exonic sites to test how this will affect SR functions. The other problem is that interpretation of data on the C-terminal nonapeptide is not accurate. This peptide region forms beta-strand in RRM1. It is not surprising that deletion or mutation of this peptide will damage structural intactness of RRM1. Similar results will be found if other regions important for structure of RRM1 is perturbed.

Additional comments

none

---

## Round 0.2 · Minor Revisions

I have taken over handling this submission as the original Academic Editor is not available.

It is understandable that any model systems (including MS2-MCP tethering assay) can have a certain degree of caveats that can be mitigated with proper controls.

While I believe no additional experiments will be needed, I would like the authors to address outstanding concerns (especially point #1) raised by reviewer 1 before acceptance of this manuscript, by discussing caveats of the MS2-MCP system, i.e., the binding of MCP alone to the MS2 site of the SMN1/2-SM2 reporters have certain effects due to steric hindrance (as discussed in the response letter). Some of the text in the response letter may be used in the main text to discuss potential caveats of the MS2 tethering assay and showcase successful examples of using it to decipher splicing factor functions despite those caveats.

Reviewer 1 ·

Basic reporting

The modifications contain a few errors in the grammar.

Experimental design

Not all the right experiments or comparisons of the data have been made. See below.

Validity of the findings

The revised manuscript contains some additional data and a small number of changes in the text inserted in response to my previous comments. The data are generally very clean, with impressively small error bars. However, I don’t think the authors quite grasped the significance of my previous comments.

1. The effects of the insertions and of binding to them by the MS2 coat protein (CP). The overwhelming effect is nothing to do with the SR proteins. Take Figures 1and S1: with the 4 constructs (SMN1 i6, SMN1 i7, SMN2 i6 and SMN2 i7) the percentages of inclusion change from 99, 95, 35 and 5, respectively, in the presence of the empty vector, to 40, 91, 1 and 41 with T7-CP. Adding SRSF1 to the T7-CP, only changes the second set to 25, 94, 1 and 31. While the SR protein made a difference, the effect is minor compared with that of just adding the MS2 CP to the T7 tag. Comparisons like this are not made in the text, and they are not included in the statistical comparisons shown in each figure. Indeed, the only reference to Figure S1 is this: “Over-expression of the four SR proteins (without CP) on SMN1/2 exon 7 splicing were firstly checked to distinguish MS2-dependent binding function (Fig. S1).” I conclude that the dominant effect is that of the MS2 CP.
Does this invalidate the work of the authors? Not completely, but Fig. S1 should be a main text figure and much more than 1 line at the start of the Results should be devoted to it.
2. Supp. Fig. 1D plays a second important role. Comparing the effects of T7 empty vector to T7-SRSF1 shows that the same set of four constructs changes in % inclusion from 99, 95, 35 and 5 to 98, 94, 42 and 12. In other words, the effect of free SRSF1 is minimal (even less than when tethered). It is really important that this finding should be set out clearly. I recommend I regard to both points that the authors should construct a single table incorporating the full results for both Fig. 1 C and D and Supp. Fig S1 b and c.
3. The missing control is a substrate in which there is neither no MS2 site (SMN wild-type) or the MS2 hairpin has been inactivated by a point mutation. This is really important if the authors wish to demonstrate that the effects mediated are via tethering. I did not see the point of Figure S10 or the text in lines 334-348 in the revised version. This is an essential control.

Reviewer 2 ·

Basic reporting

No comment

Experimental design

No comment

Validity of the findings

No comment

Additional comments

The authors have satisfactorily addressed the concerns of prior review and the revised manuscript has improved. I have no further comments.

Reviewer 3 ·

Basic reporting

The revised manuscript addressed all comments to my satisfaction.

Experimental design

The revised manuscript provided the details, like mini gene sequences and antibody used for western bloting.

Validity of the findings

Other possible interpretation of Figure 4 is provided in the discussion section now.

Additional comments

Agree to publish.

---

## Round 0.3 · accepted · Accept

The reviewers' comments have been addressed and the caveats of the MCP/MS2 tethering system for splicing studies have been discussed. The manuscript is ready for publication.